# Depolarization and lidar ratios at 355, 532, and 1064 nm and microphysical properties of aged tropospheric and stratospheric Canadian wildfire smoke

Moritz Haarig[1], Albert Ansmann[1], Holger Baars[1], Cristofer Jimenez[1], Igor Veselovskii[2], Ronny Engelmann[1], and Dietrich Althausen[1]

[1]Leibniz Institute for Tropospheric Research, Leipzig, Germany
[2]Physics Instrumentation Center of General Physics Institute, Moscow, Russia

*Correspondence to:* A. Ansmann
(albert@tropos.de)

**Abstract.**

We present spectrally resolved optical and microphysical properties of western Canadian wildfire smoke observed in a tropospheric layer from 5–6.5 km height and in a stratospheric layer from 15–16 km height during a record-breaking smoke event on 22 August 2017. Three polarization/Raman lidars were run at the European Aerosol Research Lidar Network (EARLINET) station of Leipzig, Germany, after sunset on 22 August. For the first time, the linear depolarization ratio and extinction-to-backscatter ratio (lidar ratio) of aged smoke particles was measured at all three important lidar wavelengths of 355, 532, and 1064 nm. Very different particle depolarization ratios were found in the troposphere and in the stratosphere. The obviously compact and spherical tropospheric smoke particles caused almost no depolarization of backscattered laser radiation at all three wavelength ($<$3%), whereas the dry irregularly shaped soot particles in the stratosphere lead to high depolarization ratios of 22% at 355 nm and 18% at 532 nm and a comparably low value of 4% at 1064 nm. The lidar ratios were 40-45 sr (355 nm), and 65-80 sr (532 nm), and 80-95 sr (1064 nm) in both, the tropospheric and stratospheric smoke layers indicating similar scattering and absorption properties. The strong wavelength dependence of the stratospheric depolarization ratio was probably caused by the absence of a particle coarse mode (particle mode consisting of particles with radius $>$500 nm). The stratospheric smoke particles formed a pronounced accumulation mode (in terms of particle volume or mass) centered at a particle radius of 350–400 nm. The effective particle radius was 0.32 $\mu$m. The tropospheric smoke particles were much smaller (effective radius of 0.17 $\mu$m). Mass concentrations were of the order of 5.5 $\mu$g m$^{-3}$ (tropospheric layer) and 40 $\mu$g m$^{-3}$ (stratospheric layer) in the night of 22 August 2017. The single scattering albedo of the stratospheric particles was estimated to be 0.74, 0.8, and 0.83 at 355, 532, and 1064 nm, respectively.

## 1 Introduction

A record-breaking Canadian wildfire smoke event was observed over central European lidar stations of the European Aerosol Research Lidar Network (EARLINET) on 21–22 August 2017 (Ansmann et al., 2018). Biomass-burning-smoke

was detected at almost all heights in the free troposphere and lower stratosphere from 3 to 17 km height. Intensive fires combined with rather strong pryocumulonimbus formation (Fromm et al., 2010; Peterson et al., 2017) in western Canada were most probably responsible for the optically thick stratospheric smoke layers over Europe. The arrival of the dense wildfire smoke layers over Europe in August 2017 was first reported by Khaykin et al. (2018). In our study, we present nighttime lidar observations of the extreme smoke event. Three lidars were involved in the measurements at Leipzig, Germany, after sunset on 22 August 2017. Highlight are the observations with the worldwide only triple-wavelength polarization/Raman lidar permitting the determination of the particle extinction-to-backscatter ratio (lidar ratio) and the particle linear depolarization ratio at all three important lidar wavelengths of 355, 532, and 1064 nm. Besides the spectrally resolved height profiles of smoke backscatter and extinction coefficients, lidar ratio and depolarization ratio, we discuss the microphysical properties (size distribution, volume and mass concentrations) and single-scattering albedo values of the smoke derived from the multiwavelength lidar observations, and we also contrast the findings measured in a tropospheric and the pronounced stratospheric smoke layer.

The study is an important contribution to lidar-based efforts of aerosol classification (Omar et al., 2009; Burton et al., 2012; Groß et al., 2013; Papagiannopoulos et al., 2016, 2018; Baars et al., 2017) and the establishment of a global 3-D aerosol climatology (Liu et al., 2008; Winker et al., 2010, 2013; Amiridis et al., 2015; Marinou et al., 2017). Our lidar observations add new and detailed data to the aerosol-typing library for tropospheric and stratospheric biomass-burning smoke after long-range transport over more than 10000 km. Spaceborne lidars such as CALIOP (Cloud Aerosol Lidar with Orthogonal Polarization) of NASA's CALIPSO (Cloud-Aerosol Lidar and Infrared Pathfinder Satellite Observation) mission (Winker et al., 2009; Omar et al., 2009)) and ATLID (Atmoshperic lidar) of ESA's Earth-CARE (Earth Cloud Aerosol and Radiation Explorer) mission (Illingworth et al., 2015) in combination with ground-based lidars organized in lidar networks such as EARLINET (Pappalardo et al., 2014) and the AD-Net (Asian dust and aerosol lidar observation network) (Sugimito et al., 2008, 2018) form the basis for a systematic built up a global 3-D aerosol aerosol climatology for atmospheric and climate research and future climate modeling.

One of the fundamental input parameters in the CALIOP data analysis is the particle extinction-to-backscatter ratio (lidar ratio) at 532 and 1064 nm (Omar et al., 2009; Kim et al., 2018). Observations of the lidar ratio at these two wavelengths for all important aerosol types (e.g., urban haze, marine aerosol, biomass burning smoke, desert dust) and frequently occuring mixtures of smoke pollution with mineral dust or marine particles with urban haze are a prerequisite for high-quality retrieval products and to permit accurate aerosol profiling with CALIOP on a global scale. Numerous observational lidar ratio studies are meanwhile available at 532 nm (and 355 nm, see our literature review for biomass burning aerosol in Sect. 4.2), but only recently it became possible to measure the lidar ratio at 1064 nm (Haarig et al., 2016). For the first time, we present measured 1064 nm lidar ratios for aged biomass-burning smoke.

Particle lidar-ratio and linear depolarization-ratio data sets for 355, 532, and 1064 nm for all basic aerosol types are required in efforts to harmonize long-term time series of aerosol observations with CALIOP at 532 and 1064 nm, and later on with ATLID at 355 nm. Only with good knowledge of the spectral dependencies of particle backscatter,

extinction, lidar ratio, and depolarization ratio, long-term data sets can be harmonized and normalized to permit aerosol trend analysis by combing the NASA and ESA lidar missions, spanning the time period from 2006 (start of the CALIPSO mission) to 2025 (probable end of the EarthCARE mission). Our triple-wavelength lidar observations, presented here, can be regarded as a contribution and first step towards an aerosol library containing the requested spectrally resolved aerosol optical parameters.

The paper is organized as follows: In the next section, we briefly describe the three lidars involved in the smoke study, the basic data analysis methods, and the retrieved products. In Sect. 3, the main results in terms of optical and microphysical parameters of the tropospheric and stratospheric smoke layers are presented and discussed. In Sect. 4, we make an attempt to explain the unexpected spectral dependence of the smoke linear depolarization ratio in the stratosphere. The measurement case from 22 August 2017 provides the favorable opportunity to test and improve optical models and used particle shape parameterizations. Modeling of the optical properties of irregularly shaped dust and smoke particles is a big and unsolved problem. Trustworthy parameterizations that allow us to simulate the optical properties of irregularly shaped mineral dust and soot particles at $180°$ scattering angle as function of composition (refractive index), size distribution, and especially the shape characteristics are still missing. In Sect. 4, we also present an updated literature review regarding smoke lidar ratios and depolarization ratios. EARLINET contributed significantly to this effort. Main findings and conclusions are given in Sect. 5.

## 2 Instrumentation, data analysis, and lidar products

### 2.1 Lidars

In the evening and night of 22 August 2017, three polarization/Raman lidars were run at the EARLINET station at Leipzig (51.3°N, 12.4°E, 110 m height above sea level, a.s.l.). A single-wavelength 532 nm Polly (*Po*rtab*l*e *l*idar system) (Althausen et al., 2009; Engelmann et al., 2016; Baars et al., 2016) measured the total, co-, and cross-polarized elastic backscatter signals at 532 nm, the rotational Raman signals around 532 nm, and the vibrational-rotational Raman signal at 607 nm. Co- and cross-polarized denotes here the plane of polarization with respect to the plane of the linearly polarized laser pulses. The 532 nm Polly allows us to determine height profiles of the particle backscatter coefficient, extinction coefficient, the corresponding extinction-to-backscatter ratio (lidar ratio) and the particle linear depolarization ratio at 532 nm. Specific details to the data analysis are given in Sect. 2.2.

The second lidar was the dual receiver field-of-view (RFOV) multiwavelength polarization/Raman lidar MARTHA (Multi-wavelength Tropospheric Raman lidar for Temperature, Humidity, and Aerosol profiling) (Mattis et al., 2003, 2008; Schmidt et al., 2013, 2014; Jimenez et al., 2017). MARTHA has a powerful laser transmitting in total 1 J per pulse at a repetition rate of 30 Hz and has an 80 cm telescope, and is thus well designed for tropospheric and stratospheric aerosol observations (Mattis et al., 2004, 2008, 2010). This lidar measures Raman signals at 532 and 607 nm and polarization-sensitive 532 nm backscatter signals at two RFOVs so that besides aerosol profiles, cloud microphysical properties can be retrieved from measured cloud multiple scattering effects. We used the 532 nm particle depolarization ratio measured with the smaller RFOV in the study presented

here. Furthermore, the 355, 532, and 1064 nm particle backscatter coefficients, the 355 and 532 nm extinction coefficient profiles and the corresponding lidar ratio profiles are presented in the result section.

The third Leipzig lidar is the triple-wavelength polarization/Raman lidar BERTHA (Backscatter Extinction lidar-Ratio Temperature Humidity profiling Apparatus) (Althausen et al., 2000; Tesche et al., 2011; Haarig et al., 2016, 2017a). BERTHA was

5 designed and optimized for desert dust characterization and participated in a series of dust field campaigns, e.g., at Barbados in 2013 and 2014 in the framework of the Saharan Aerosol Long-Range Transport and Aerosol–Cloud-interaction Experiment SALTRACE (Weinzierl et al., 2017; Haarig et al., 2017a). BERTHA allows us to measure particle linear depolarization ratios and lidar ratios at all three important lidar wavelengths of 355, 532, and 1064 nm. In the present configuration, the 1064 nm depolarization ratio and the 1064 nm lidar ratio can only be measured alternatively (not simultaneously). The 1064 nm depo-

10 larization sensitive channel (cross-polarized channel) can be substituted by a 1058 nm rotational Raman channel within 20-30 minutes. This procedure includes adjustment and signal optimizing efforts. On 22 August 2017, we first measured the 1058 nm Raman signal profiles (for 2.5 hours) to obtain the 1064 nm extinction profile, and afterwards the cross-polarized 1064 nm signal component (for 40 minutes), needed in the retrieval of the 1064 nm depolarization ratio.

The laser beams of Polly and BERTHA were tilted to an off-zenith angle of 5° in different directions, whereas MARTHA

was pointing to the zenith, which leads to horizontal distances between the laser beams of the order of 450-750 m at the base of the tropospheric smoke layer at 5 km height and of 1.3-2 km at the base of the stratospheric layer at 15 km height. However, the good agreement of the results as discussed in Sect. 3 indicated that the smoke layers were obviously horizontally homogeneous on scales of 1–2 km.

## 2.2 Lidar data analysis: optical properties

Details of the determination of the particle optical properties and the uncertainties in the products can be found in the articles mentioned above. An overview of the retrieval methods is given in Ansmann and Müller (2005); Freudenthaler et al. (2009), and Freudenthaler (2016). The Raman lidar method was exclusively used to determine particle backscatter and extinction profiles. The particle backscatter coefficient is obtained from the measured ratio of the elastic backscatter signal to the respective Raman signal. The 355 nm and 532 nm particle extinction coefficients are computed from the vibrational-rotational Raman

signals measured at 387 and 607 nm, respectively. The 1064 nm extinction coefficients are calculated from rotational Raman signals measured around 1058 nm. In the correction of Rayleigh extinction and backscattering effects, temperature and pressure profiles from the GDAS (Global Data Assimilation System) data base are used (GDAS, 2018). The determination of the particle linear depolarization ratio from the volume depolarization ratio, shown in Ansmann et al. (2018), is described in detail by Freudenthaler et al. (2009); Freudenthaler (2016), and Haarig et al. (2017a). The linear depolarization ratio is defined as the

cross-polarized-to-co-polarized backscatter ratio. Co and cross denote the planes of polarization parallel and orthogonal to the plane of linear polarization of the transmitted laser pulses, respectively.

In Sect. 3, the lidar results for the time period from 20:45 – 23:15 UTC on 22 August 2017 (shown in Fig. 1 in Sect. 3) are presented and discussed. The lidar signals for the selected time period of 150 minutes were averaged and background-corrected before the optical properties (backscatter and extinction coefficients, depolarization ratios) were computed. This procedure was

performed separately and independently for all three lidar data sets. In case of the 1064 nm depolarization ratio observations with BERTHA, we averaged the signals from 23:50 – 00:30 UTC. The signal profiles had to be smoothed afterwards to reduce the impact of signal noise to a tolerable level on the final products. In the case of the backscatter coefficients and the particle depolarization ratio (determined from the measured profiles of the cross-polarized and total (cross- + co-polarized) elastic backscatter signal components), we smoothed the individual signal profiles with vertical gliding averaging window lengths of 50-100 m (backscatter coefficients, troposphere), 100-250 m (backscatter coefficient, stratosphere), and 200-400 m (depolarization ratio, troposphere and stratosphere).

In the retrieval of the extinction coefficient, a least-squares linear regression method was applied to the respective Raman signal profiles. The regression window length was 750 m (532 nm) to 1200 m (355 nm) in the troposphere and 1200 m for both wavelengths in the stratosphere. To obtain the lidar ratios at 355 and 532 nm, the extinction profiles were combined with the respective backscatter profiles. In this procedure, we applied the optimum-effective-resolution concept (Iarlori et al., 2015; Mattis et al., 2016) and used a smoothing window length in the backscatter retrieval which was around 0.78 of the regression window length in the extinction retrieval.

In the case of the 1064 nm extinction coefficient, only smoke-layer mean extinction values could be derived. Profiles could not be obtained because the 1058 nm Raman signals were too weak and noisy. The retrieval window lengths are indicated by vertical bars in the figures in the result section (Sect. 3). Retrieval window lengths of 750–1500 m in the troposphere and 2500 m in the stratosphere had to be applied to obtain the 1064 nm layer-mean extinction coefficient with an uncertainty of about 10% (troposphere) and <50% (stratosphere). The retrieval of the 1064 nm layer-mean lidar ratio for the observed pronounced smoke layers is explained in Sect. 3, when the results in the respective figures are described.

Different expressions for the Ångström exponent, a well-established parameter to characterize the spectral dependence of aerosol optical properties, are shown in Sect. 3. The Ångström exponent $a_{x,\lambda_1/\lambda_2} = \ln(x_1/x_2)/\ln(\lambda_2/\lambda_1)$ describe the wavelength dependence of an optical parameter $x$ in the spectral range from wavelength $\lambda_1$ to $\lambda_2$. $x_i$ may be the backscatter coefficient ($x_i = \beta(\lambda_i)$) or the extinction coefficient ($x_i = \sigma(\lambda_i)$) or the lidar ratio ($x_i = S(\lambda_i)$). The following relationship holds between the backscatter-related, extinction-related and lidar-ratio-related Ångström exponent: $a_{\sigma,\lambda_1/\lambda_2} = a_{\beta,\lambda_1/\lambda_2} + a_{S,\lambda_1/\lambda_2}$ (Ansmann et al., 2002).

## 2.3 Lidar data analysis: microphysical properties

The lidar inversion method of Veselovskii et al. (2002, 2010) is applied to obtain microphysical particle properties such as the particle effective radius, volume and surface area concentrations, size distribution, and refractive index characteristics from the measured optical properties, i.e., from particle backscatter coefficients at 355, 532, and 1064 nm and extinction coefficients at 355 and 532 nm. The data analysis assumes spherical smoke particles in the tropospheric layer and 10-15% uncertainty in the measured backscatter and extinction profiles. In the retrieval of the microphysical properties of stratospheric smoke, spherical as well as spheroidal particles are assumed. The single scattering albedo (SSA) of the smoke particles, presented in Sect. 3, is computed from the retrieved particle size distribution and the most appropriate refractive index characteristics (real and imaginary parts) used as input in the lidar inversion procedure.

# 3 Observations

## 3.1 Overview

The record-breaking Canadian wildfire smoke event over Leipzig on 22 August 2017 was discussed by Ansmann et al. (2018). It was shown that the total (tropospheric+stratospheric) smoke-related AOT at 532 nm reached values close to 1.0 during the noon hours. Smoke was present at all heights in the free troposphere as well as in the lower stratosphere over central Europe up to 17 km height. An optically dense stratospheric layer extended from 14–16 km height over Leipzig and showed a 532 nm AOT of 0.6. **As discussed by Ansmann et al. (2018) and Hu et al. (2018), record-breaking intensive fires combined with the formation of exceptionally strong pyrocumulonimbus clusters in the southern parts of British Columbia in western Canada and the northwestern United States in the afternoon of 12 August 2017 were most probably responsible for these unprecedentedly optically thick stratospheric smoke layers reaching Europe. Within cumulus towers enormous amounts of smoke can be injected into the upper troposphere and lower stratosphere (Fromm et al., 2003, 2010; Rosenfeld et al., 2007; Peterson et al., 2017). Because the lifting is so fast (from the fire sources at ground to the upper troposphere and lower stratosphere within $<$1 hour in convective cloud systems), most of the smoke particles reach the tropopause region without any interaction with trace gases, other aerosol particles, and cloud drops. The majority of the stratospheric smoke particles on 22 August 2017 was obviously uncoated, pure, irregularly shaped soot particles as our observations discussed below corroborate.**

Figure 1 shows the aerosol layering over Leipzig in the night of 22 August 2017, about 10 hours after the maximum stratospheric contamination observed over Leipzig. Tropospheric aerosol layers were present from the surface to about 6.5–7 km height. The top of the planetary boundary layer (PBL, residual layer during nighttime) was around 1.8 km height, as the enhanced range-corrected signal (red color) around 1.8 km height before 22:00 UTC in Fig. 1 indicates. Between 8 and 13-14 km height, the atmosphere was almost free of smoke. The main stratospheric smoke layer is visible between 15 and 16 km height. The stratospheric layer was about 3–4 km above the tropopause. The 532 nm AOT of the stratospheric layer had decreased from 0.6 around noon to 0.2–0.25 in the night of 22 August 2017. In the following two subsection, the basic lidar results are presented.

## 3.2 Smoke optical properties

In Figs. 2 and 3, the results of the observations with three polarization/Raman lidars are presented. Mean height profiles of the optical properties for the time period from 20:45–23:15 UTC on 22 August 2017 are shown, except for the 1064 nm depolarization ratio (23:50–0:30 UTC, see Fig. 1, and explanations in Sect. 2.2). Figure 2 shows the smoke optical properties in the tropospheric layer. Figure 3 contains the respective findings for the stratospheric smoke layer. Table 1 provides an overview of layer-mean values of smoke optical properties for the pronounced tropopsheric layer from 5–6.5 km height and the stratospheric layer from 15–16 km height. Table 1 considers all profiles shown in Figs. 2 and 3 obtained with the three lidars. According to the backward trajectory analysis presented by Ansmann et al. (2018), the wildfire smoke traveled about 7–10 days from the fire sources in western Canada to central Europe.

As can be seen in Figs. 2 and 3, a good agreement between the observations with BERTHA, MARTHA, and Polly is given for all parameters. However, a high impact of signal noise on the retrieved profiles is visible as well. The MARTHA 355 nm extinction profile could be measured up to about 15.3 km only. The high signal noise is due to the fact that we avoided any overloading of the photomultipliers (operated in the photon counting mode) so that even the strong near-range signals in the lowest part of the troposphere were properly measured. As a consequence, the signals were comparably weak in the middle and upper troposphere and lower stratosphere and therefore the influence of signal noise likewise high. This measurement strategy was selected to obtain reliable backscatter and extinction profiles almost from the ground to the top of the stratospheric smoke layer so that the full extinction profiles (as well as the integral) is available for comparison with AERONET sun photometer observations.

**In the case of the 1064 nm extinction coefficient, we only can show a few values in Figs. 2 and 3. The retrieval window lengths are indicated by vertical bars. In the case of stratospheric smoke, a regression window length of 2500 m was required to obtain the 1064 nm extinction coefficient with a statistical uncertainty of about 50%. The vertical extent of the stratospheric layer was, however, 1250 m only. The shown 1064 nm extinction coefficient of 62.5 $Mm^{-1}$ in Fig. 3 is the mean value for the vertical regression-fit interval of 2500 m. The 1064 nm AOT of this 2500 m thick layer is 0.156. This AOT of 0.156 combined with the true geometrical depth of the stratospheric layer of 1250 m yields the layer mean particle extinction coefficient of 125 $Mm^{-1}$. This extinction value is given in Table 1 and the basis for the calculation of the 1064 nm lidar ratio shown in Fig. 3 and given in Table 1. To obtain the 1064 nm lidar ratio, we combined the extinction value for the 1250 m layer with the respective backscatter coefficient computed from signal profiles smoothed with a window length of 937.5m vertical window length according to the effective resolution concept (Iarlori et al., 2015; Mattis et al., 2016).**

The key findings shown in Figs. 2 and 3 and listed in Table 1 can be summarized as follows. As already observed during previous Canadian wildfire events (see the literature review in Sect. 4.2), the tropospheric backscatter coefficient of aged wildfire smoke shows a clear and strong wavelength dependence for the 355-532 nm wavelength range, whereas the wavelength dependence of the respective extinction coefficient in the 355-532 nm spectral range is much weaker. Consequently, the 532 nm lidar ratio for aged smoke is larger than the 355 nm lidar ratio. For urban haze and fresh smoke, the 355 nm lidar ratio is typically larger than the 532 nm lidar ratio. The wavelength dependencies are reflected in the shown profiles for the backscatter and extinction-related Ångström exponents in Figs. 2d and 3d. The Ångström exponents (for the 355–532 nm spectrum) are higher in the troposphere than in the stratosphere. This may indicate that the stratospheric smoke particles were larger (see next subsection). The comparably high stratospheric backscatter Ångström exponent (for the 532–1064 nm wavelength region) in Fig. 3 is indicative for the absence of coarse smoke particles, i.e., particles with diameter $>1$ $\mu$m. This aspect is further illuminated in the next subsection.

The most surprising finding is the strong difference between the depolarization spectrum in the tropospheric and stratospheric smoke layers as shown in Figs. 2e and 3e. The depolarization ratios were below 3% for all three wavelengths in the tropospheric smoke layer (seen by all three lidars). This is a clear indication that the particles were (almost) spherical in shape, e.g., soot particles coated with liquid material (Dahlkötter et al., 2014). In strong contrast, high depolarization ratios of 22% and 18%

were observed at 355 and 532 nm, respectively, in the stratosphere. The depolarization ratios was again low (4%) at 1064 nm. Strong depolarization of the transmitted linearly polarized laser radiation points to irregularly shaped particles. The unexpected strong wavelength dependence of the particle linear depolarization ratio in the stratospheric layer and possible reasons for this spectral behavior is discussed in Sect. 4.2.

### 3.3 Smoke microphysical properties

Table 2 provides an overview of the microphysical properties of the aged biomass-burning smoke in the tropospheric and stratopspheric aerosol layer. The microphysical properties are obtained by applying the lidar inversion method described in Sect. 2.3 to the extinction coefficients at 355 and 532 nm in Table 1 and the corresponding backscatter coefficients at 355, 532, and 1064 nm computed from the extinction coefficients, lidar ratios, and respective Ångström exponents in Table 1. The particle mass concentrations were then computed from the volume concentrations by assuming a smoke particle density of 1.35 g cm$^{-3}$ (Reid and Hobbs, 1998).

As can be seen in Table 2, the aerosol load was much larger in the stratospheric layer. Mass concentrations were 5–6 $\mu$g m$^{-3}$ in the tropospheric layer and close to 40 $\mu$g m$^{-3}$ in the stratospheric layer at the nighttime hours. A clear indication for the presence of highly absorbing stratospheric particles is the low SSA of 0.80–0.85 at 532 nm and 1064 nm. Around noon of 22 August 2017, the stratospheric smoke particle number concentration, volume and mass concentrations were about a factor of two higher than the values in Table 2. Peak mass concentrations in the stratosphere reached values of 70–100 $\mu$g m$^{-3}$ (Ansmann et al., 2018).

Figure 4 shows the particle mass size distributions retrieved by means of the lidar data inversion analysis. The size distribution for the particle mass concentration is obtained by multiplying the derived volume size distribution with the smoke particle density of 1.35 g cm$^{-3}$. The respective particle mass size distribution derived from the AERONET observation at Lindenberg in the morning of 23 August 2017 is shown for comparison. The AERONET size distributions were downloaded from the AERONET data base (AERONET, 2018). A brief discussion of the AERONET products (Holben et al., 1998) and retrieval methods can be found in Ansmann et al. (2018).

The AERONET observation describes the aerosol properties in the entire vertical column from the surface to the top of the stratopspheric layer. To convert the AERONET column values to stratospheric volume and mass concentrations so that we can compare sun-photometer-derived and lidar-derived stratospheric volume and mass concentrations, we assumed that (a) the stratospheric smoke contributed 60% to the total AOT (as observed with lidar) and also 60% to the column volume concentration, and (b) that these 60% can be assigned to the 1 km thick stratospheric layer from 15 and 16 km height. With this information, the AERONET column volume values for each size bin were converted into volume and mass concentrations as shown in Fig. 4 and interpreted as the stratospheric contribution to the total column mass size distribution.

The lidar-derived size distribution (from the nighttime measurements) fits very well into the AERONET observations at Lindenberg, 180 km northeast of Leipzig, in the early morning of 23 August 2017. The effective radius $r_{\text{eff}}$ of the smoke particles in the stratospheric layer was 0.32 $\mu$m (see Table 2). According to the AERONET observations at Lindenberg, the total effective radius $r_{\text{eff}}$ (for the entire particle size distribution and for the entire vertical column) was 0.33–0.42 $\mu$m and

the fine-mode effective radius $r_{\mathrm{eff,f}}$ controlled by urban haze in the boundary layer and the fire smoke in the free troposphere and lower stratosphere was 0.23–0.32 $\mu$m. Coarse mode particles (particles with radius >500 nm) were almost absent. The remaining weak impact of coarse particles on the volume size distribution for the entire vertical column is probably related to

the occurrence of soil and road dust in the boundary layer over Lindenberg.

     The lidar-derived and AERONET-derived mass size distributions in Fig. 4 provide a consistent picture of the smoke-related tropospheric and stratospheric size distributions. The pronounced accumulation mode in the AERONET column observation is clearly caused by stratospheric smoke particles. By comparing the tropospheric and stratospheric size distributions we see that the particles were small in the tropospheric smoke layer. The size distribution in the stratosphere in Fig. 4 is in good agreement

with airborne in situ smoke observations. Similar size distributions with a pronounced smoke accumulation mode shifted to larger sizes were found during several airborne in situ measurements of North Amercian wildfire plumes in the European region (Fiebig et al., 2002; Petzold et al., 2007; Dahlkötter et al., 2014). All these airborne in situ observations indicated that super micrometer particles (coarse–mode particles) were almost absent in the aged smoke plumes and that the accumulation mode was enhanced and shifted towards larger mode diameters.

The lidar inversion results in Table 2 and the size distribution in Fig. 4 do not change much when assuming spheroidal instead of spherical particles in the lidar inversion procedure. We hypothesize that the reason for the low impact of particle shape on the retrieval products is the absence of a particle coarse mode in the stratospheric smoke layer so that the particles were at all likewise small. At these conditions shape aspects have a low impact on the lidar inversion products.

## 4   Discussion

**4.1   The unexpected wavelength dependence of the stratospheric depolarization ratio: an attempt of explanation**

Figure 5 highlights the most surprising finding. The smoke layer mean depolarization and lidar ratio values obtained from the BERTHA observations in Figs. 2 and 3 are shown. Very different depolarization spectra were found in the troposphere and stratosphere (see Fig. 5b). In contrast, the lidar ratios showed quite similar values and a similar wavelength dependence in both layers (Fig. 5a). The same origin of the aerosol and thus similar aerosol composition resulting in similar basic scattering and

absorption properties may be the reason for the less variable lidar ratios in the tropospheric and stratospheric layers.

     As already mentioned in Sect. 3.2, the particle depolarization ratio was low at all three wavelengths in the tropospheric layer. These low depolarization values are indicative for spherical particles dominating the measured optical effects. The particles must have been compact in shape (China et al., 2015). Many of them may have been composed of a solid soot core with liquid sulfate shell (Zhang et al., 2008; Adachi et al., 2010; Dahlkötter et al., 2014; China et al., 2015). Some irregularly shaped soil

dust particles (Nisantzi et al., 2014) together with partly coated soot particles may have caused the slightly enhanced 355 and 532 nm depolarization ratios. Stratospheric soot particles, on the other hand side, can be rather irregularly in shape as observed and shown (photographs) by Strawa et al. (1999). We hypothesized in Sect. 3.1 that the fast vertical transport of the fire smoke to the upper troposphere and lower stratosphere by pyroconvection may have prohibited interaction with gases (and coating)

and internal mixing with other aerosol particles so that pure irregularly shaped soot particles could enter the rather dry lower stratosphere.

The strong spectral slope of the depolarization ratio of the stratospheric smoke particles was also measured with a triple-wavelength polarization lidar at Lille, northern France, performed in Canadian wild fire smoke layers from 24–31 August 2017 (Hu et al., 2018). Values of 23–28% (355 nm), 18–20% (532 nm), and 4–5% (1064 nm) were observed. Furthermore, Burton et al. (2015) found a rather similar wavelength dependence of the depolarization ratio in a well-defined layer of wildfire smoke advected from the Pacific Northwest of the United States to the Boulder-Denver region at 8 km height. The depolarization ratio decreased from 21% (355 nm) to 9% (532 nm) and 1% (1064 nm) in 8 km height.

The question arises: What is the reason for this unexpectedly strong spectral dependence of the particle linear depolarization ratio in the stratosphere? Usually, the observed wavelength dependence of the depolarization ratio of irregularly shaped particles (such as mineral dust or volcanic ash) is weak. The depolarization ratio for desert dust is lower at 355 nm (20–25%) than at 532 nm (30–35%) (Groß et al., 2013, 2015; Burton et al., 2015; Hofer et al., 2017; Haarig et al., 2017a), and decreases again towards 1064 nm (20–25%) (Burton et al., 2015; Haarig et al., 2017a). Close to dust sources, depolarization ratios were equal at 532 and 1064 nm were equal with values close to 40% (Burton et al., 2015). Particle depolarization ratios exceeded 50% during the passage of dust devils containg a considerable amount of giant dust and even sand particles (per definition particles with radius >30 $\mu$m) (Ansmann et al., 2009).

We hypothesize that the specific size distribution of the stratospheric smoke particles shown in Fig. 4 is responsible for the strong wavelength dependence. A pronounced accumulation prevailed and coarse mode particles were absent. In the case of typical dust plumes (close to the desert source regions and even after long range transport) the coarse mode is dominating and depolarization of linearly polarized laser radiation is strong at all three wavelengths. Most of the optically active dust particles are large compared to the three laser wavelengths. The larger the particles for a given wavelength, the larger the linear depolarization ratio (Gasteiger et al., 2011). In the case of a pronounced accumulation mode, the particles are still large for the wavelength of 355 nm, but small for the wavelength of 1064 nm. The missing smoke coarse mode thus explains to our opinion the strong wavelength dependence of the smoke depolarization ratio.

**Our hypothesis is corroborated by Fig. 6. We compare the spectrum of the linear depolarization ratio of stratospheric smoke in Fig. 5 with the depolarization ratio spectrum for fine-mode desert dust as observed in laboratory studies of Järvinen et al. (2016) (see review of Mamouri and Ansmann (2017) for more details). In these laboratory studies, well defined monomodal size distribution of dust particles were produced (from fine-mode to coarse-mode size distributions) and the respective particle linear depolarization ratios of the dust particle ensembles were measured as a function of size (mode radius). Based on these studies, Mamouri and Ansmann (2017) concluded that fine-mode dust causes depolarization ratios of 20–22% (355 nm), 15-16% (532 nm) and <10% (1064 nm). These values are shown in Fig. 6.**

**As can be seen, quite similar values of the soot and fine-mode dust depolarization ratios at 355 and 532 nm are found. The rather different composition of the particles (soot vs dust particles) is not visible in the two depolarization spectra. Obviously particle shape and size widely control the strength of depolarization of backscattered laser radiation at the different wavelengths.**

However, to obtain clear answers concerning the role of particle size, shape and composition on light depolarization, we need extended simulation studies with advanced optical particle models (Gasteiger et al., 2011; Lindqvist et al., 2014; Kemppinen et al., 2015a, b; Mishchenko et al., 2016). Unfortunately, modelling of the optical properties of dust and soot particles for our specific lidar application (scattering at exactly $180°$) is a very crucial task. A satisfactory reproduction of measured lidar ratio and depolarization ratio spectral behavior for irregularly shaped dust and smoke particles and even for cubic-like sea salt particles (Haarig et al., 2017b) is hard to achieve. Our unique observations of lidar ratios and depolarization ratios for irregularly shaped soot particles at 355, 532, and 1064 nm can be regarded as a favorable test case to improve especially the particle shape parameterizations in optical particle models.

## 4.2 Smoke depolarization and lidar ratios: an updated literature review

To compare our lidar and depolarization ratio observations in Sect. 3.1 with previous lidar observations of wildfire smoke, we performed a literature review. Numerous articles on tropospheric biomass-burning smoke are available. In Table 3, we provide an overview of our literature review regarding multiwavelength lidar observations of fresh and aged smoke. It was already noticed more than 10 years ago that the 355 nm lidar ratio for aged smoke after many days of long-range transport is considerably lower than the 532 nm lidar ratio. This is consistent with the discussion and the findings presented above. As can be seen in Table 3, the difference between the 355 and 532 nm lidar ratios can be as large as 15–25 sr. For fresh smoke, advected from fire sources to the lidar stations within less than 2–3 days, the lidar ratios at 355 and 532 nm are similar or the 355 nm values are larger.

Only a few observations of the particle depolarization ratio in aged and fresh tropospheric smoke layers are available as can be seen in Table 3. The low values are indicative for small and spherical smoke particles and moderately increased depolarization ratio may be related to the presence of some soil dust in the smoke plumes or may be caused by the presence of partly coated smoke particles. The high depolarization ratios at 355 and 532 nm and the strong wavelength dependence of the depolarization ratio as observed in the August 2017 stratospheric smoke layers are a relatively new features and were first observed in an elevated aged smoke layer in the upper troposphere by Burton et al. (2015).

## 5 Conclusions

A record-breaking stratospheric smoke event with aerosol layers from tropospheric heights around 3 km to about 16-17 km height allowed us to characterize Canadian wildfire smoke after long-range transport in large detail in terms of optical and microphysical properties. The case study demonstrates the unique potential of advanced aerosol lidars to contribute to atmospheric aerosol research. There is no alternative to lidar regarding a continuous aerosol profiling over long time periods providing a clear separation of tropospheric and stratospheric aerosol effects. Our worldwide only triple-wavelength polarization/Raman lidar delivered height profiles of particle backscatter and extinction coefficients, respective lidar ratios, and linear depolarization ratios at all three lidar wavelengths, and, in addition, microphysical, morphological, and composition-related information about the smoke layers. Very different smoke properties were

observed in the tropospheric and stratospheric smoke layers. For the first time, measured lidar ratios for smoke at 1064 nm are now available, and can be used in the CALIOP data analysis of the spread of the smoke over the northern hemisphere occurring during the second half year of 2017 (Khaykin et al., 2018).

The spectrally resolved optical data sets for stratospheric smoke is an important new contribution to the aerosol-typing library used in lidar remote sensing from space and by ground-based networks such as EARLINET. The smoke observations also provide a favorable opportunity to test and validate optical models regarding their potential to reproduce the observed data sets of smoke lidar and depolarization ratios. Improved modeling will in turn help to better interpret aerosol lidar observations and support the development of new lidar retrieval algorithms, and also to improve climate modeling by using improved optical aerosol models for non-spherical particles.

It should finally be mentioned that the lidar-derived optical properties for stratospheric smoke are rather different from the ones of non-absorbing and non-depolarizing, spherical volcanic aerosol particles (liquid sulfuric-acid containing droplets). A clear and unambiguous discrimination of biomass burning smoke and volcanic aerosol is possible based on polarization lidar observations and thus a clear and unambiguous identification of these major contributors to stratospheric aerosol perturbations and contamination.

As an outlook, we plan to use the triple-wavelength polarization/Raman lidar to characterize other fundamental aerosol types such as desert dust and urban haze in terms of lidar ratios and depolarization ratios at 355, 532, and 1064 nm to support the CALIOP observations and future spaceborne missions and space lidar data harmonization efforts. First attempts in pure marine environments and in lofted mineral dust layers have been published (Haarig et al., 2017a, b) although lidar ratios at 1064 nm could only be estimated from combined AERONET sun photometer and aerosol lidar observations, but not directly measured, to that time.

## 6  Data availability

The lidar data are available at TROPOS upon request (info@tropos.de). AERONET sun photometer data are downloaded from the AERONET web page (AERONET, 2018).

*Acknowledgements.* We are grateful to AERONET for providing high quality sun photometer observations, calibrations, and products. Special thanks to the Lindenberg AERONET team to carefully run the station. This activity is supported by ACTRIS Research Infrastructure (EU H2020-R&I) under grant agreement no. 654109. The development of the lidar inversion algorithm was supported by the Russian Science Foundation (project 16-17-10241).

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

**Table 1.** Optical properties of smoke aerosol in the tropospheric layer (5–6.5 km height) and stratospheric smoke layer (15-16 km height). Layer mean values of the particle extinction coefficient $\sigma$, lidar ratio $S$, particle linear depolarization ratio $\delta$, and backscatter-related and extinction-related Ångström exponent $a_{\sigma,\lambda_1/\lambda_2}$ and $a_{\beta,\lambda_1/\lambda_2}$ for the wavelength range from $\lambda_1$ to $\lambda_2$ are given. The Ångström exponents are directly computed from the particle backscatter and extinction values in this table. The listed layer mean values (and retrieval uncertainties) are based on all available observations (with all three lidars) taken in the night of 22 August 2017.

| Parameter | Troposphere | | | Stratosphere | | |
|---|---|---|---|---|---|---|
| | 355 nm | 532 nm | 1064 nm | 355 nm | 532 nm | 1064 nm |
| $\sigma$ | $60\pm6$ Mm$^{-1}$ | $42\pm3$ Mm$^{-1}$ | $28\pm7$ Mm$^{-1}$ | $200\pm16$ Mm$^{-1}$ | $225\pm13$ Mm$^{-1}$ | $125\pm33$ Mm$^{-1}$ |
| $S$ | $45\pm5$ sr | $68\pm9$ sr | $82\pm27$ sr | $40\pm4$ sr | $72\pm9$ sr | $92\pm28$ sr |
| $\delta$ | $2\pm4\%$ | $3\pm2\%$ | $1\pm0.8\%$ | $22\pm2\%$ | $18\pm1\%$ | $4\pm0.8\%$ |
| $a_{\sigma,355/532}$ | $0.9\pm0.5$ | | | $-0.3\pm0.4$ | | |
| $a_{\sigma,532/1064}$ | | $0.6\pm0.3$ | | | $0.85\pm0.3$ | |
| $a_{\beta,355/532}$ | $2.1\pm0.6$ | | | $1.2\pm0.4$ | | |
| $a_{\beta,532/1064}$ | | $0.8\pm0.3$ | | | $1.2\pm0.6$ | |

**Table 2.** Lidar inversion products (assuming spherical particles) for the tropospheric layer (5–6.5 km height) and stratospheric smoke layer (15-16 km height). Layer mean values (and retrieval uncertainties) of the particle volume concentration $V$, mass concentration $m$, effective radius $r_{\mathrm{eff}}$, number concentration $N$, and single scattering albedo SSA are given. The lidar inversion analysis considers 10–15% uncertainty in the measured optical properties.

| Parameter | Troposphere | Stratosphere |
|---|---|---|
| $V$ | $4\pm1.2$ $\mu$m$^3$ cm$^{-3}$ | $28\pm9$ $\mu$m$^3$ cm$^{-3}$ |
| $m$ | $5.5\pm1.8$ $\mu$g m$^{-3}$ | $38\pm12$ $\mu$g m$^{-3}$ |
| $r_{\mathrm{eff}}$ | $0.17\pm0.06$ $\mu$m | $0.32\pm0.10$ $\mu$m |
| $N$ | $212\pm80$ cm$^{-3}$ | $323\pm120$ cm$^{-3}$ |
| SSA | | $0.74\pm0.05$ (355 nm) |
| | | $0.80\pm0.05$ (532 nm) |
| | | $0.83\pm0.05$ (1064 nm) |

**Table 3.** Literature overview of multiwavelength lidar observations of smoke lidar ratios and particle linear depolarization ratios of fresh and aged biomass-burning smoke in the troposphere and stratosphere. For better comparison, the tropospheric triple-wavelength depolarization ratio observation of Burton et al. (2015) performed in aged northwestern American smoke is listed in the third line, i.e., before the tropospheric section. The stratospheric lidar and depolarization ratios of Hu et al. (2018) were measured at Lille, northern France, on, 24–31 August 2017. The range of mean values of 6 lidar sessions at Lille (on 4 different days) is given, retrieval uncertainties are of the order of 10-20 sr for the lidar ratios and 0.01 (1064 nm), and 0.03–0.04 (355 and 532 nm) in the case of the depolarization ratios.

| Study | Lidar ratio | | | Depolarization ratio | | |
|---|---|---|---|---|---|---|
| | 355 nm | 532 nm | 1064 nm | 355 nm | 532 nm | 1064 nm |
| **Stratosphere, Canadian smoke** | | | | | | |
| This study, aged | 40±16 sr | 66±12 sr | 92±27 sr | 0.224±0.015 | 0.184±0.006 | 0.043±0.007 |
| Hu et al. (2018), aged | 31-45 sr | 54-58 sr | – | 0.23-0.28 | 0.18-0.20 | 0.04-0.05 |
| Burton et al. (2015), aged | – | – | – | 0.203±0.036 | 0.093±0.015 | 0.018±0.002 |
| **Troposphere, Canadian and Siberian smoke** | | | | | | |
| Wandinger et al. (2002) and | | | | | | |
| Fiebig et al. (2002), aged | 40–70 sr | 40–80 sr | – | – | 0.06–0.11 | – |
| Murayama et al. (2004), aged | 40 sr | 65 sr | – | – | 0.06 | – |
| Müller et al. (2005), aged | 30–55 sr | 40–60 sr | – | – | – | – |
| Veselovskii et al. (2015), fresh | 65–90 sr | 65–80 sr | – | – | – | – |
| Ortiz-Amezcua et al. (2017), aged | 23–34 sr | 47–58 sr | – | – | 0.02–0.08 | – |
| Janicka et al. (2017), aged | 60±20 sr | 100±30 sr | – | 0.01–0.05 | 0.02–0.04 | – |
| This study, aged | 46±6 sr | 67±4 sr | 82±22 sr | 0.021±0.040 | 0.029±0.015 | 0.009±0.008 |
| **Troposphere, European smoke** | | | | | | |
| Alados-Arboledas et al. (2011) | 60–65 sr | 60–65 sr | – | – | – | – |
| Nicolae et al. (2013) | 73±12 sr | 46±6 sr | – | – | – | – |
| Nicolae et al. (2013), aged | 40±16 sr | 54±10 | – | – | – | – |
| Pereira et al. (2014) | 56±6 sr | 56±6 sr | – | – | 0.05±0.01 | – |
| **Troposphere, Amazonian smoke** | | | | | | |
| Baars et al. (2012), aged | 62±12 sr | 64±15 sr | – | 0.025±0.01 | – | – |
| **Troposphere, African smoke** | | | | | | |
| Tesche et al. (2011) | 87±17 sr | 79±17 sr | – | – | – | – |
| Giannakaki et al. (2015) | 89±20 sr | 83±23 sr | – | – | – | – |

**Figure 1.** Canadian wildfire smoke layers in the troposphere (mostly between boundary-layer top at 1.8 and 6.5 km height) and in the stratosphere (15–16 km height) observed with lidar at Leipzig on 22-23 August 2017, 20:45–00:30 UTC. Shown is the range-corrected cross-polarized 532 nm backscatter signal measured with temporal and vertical resolution of 10 s and 7.5 m, respectively. The indicated tropopause height (GDAS, 2018) is in agreement with nearby radiosonde profiles.

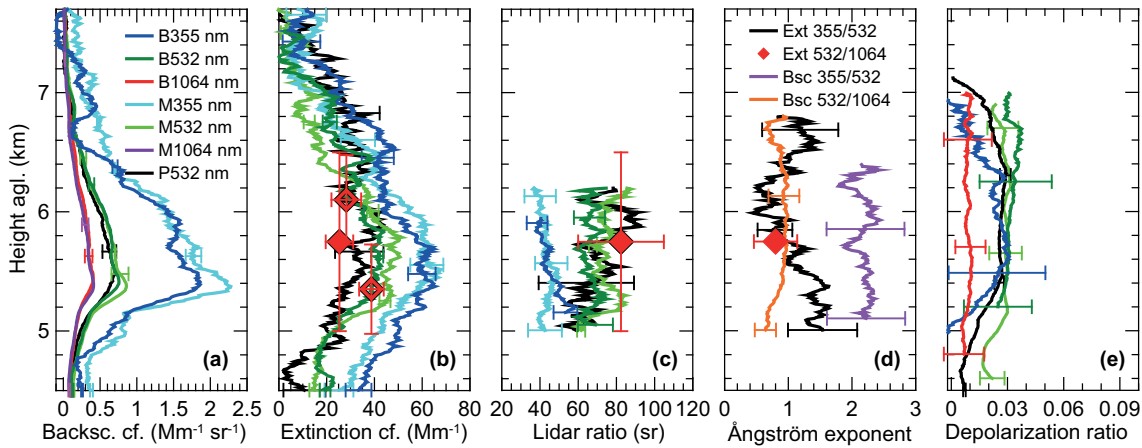

**Figure 2.** 2.5-hour mean profiles (20:45-23:15 UTC, see Fig. 1) of optical properties in the tropospheric smoke layer: (a) particle backscatter coefficient at three wavelengths measured with three lidars (BERTHA, B, MARTHA, M, and 532 nm Polly, P), (b) respective extinction coefficients (colors as in a), (c) extinction-to-backscatter ratio (lidar ratio, colors as in a), (d) backscatter-related (Bsc) and extinction-related (Ext) Ångström exponents (BERTHA only), and (e) particle linear depolarization ratio (colors as in a). Error bars indicate the retrieval uncertainty (one standard deviation). In the case of the 1064 nm extinction coefficient, a height profile could not be determined. Therefore, only a few values for retrieval window length (least-squares method) of 750 and 1500 m (indicated by vertical bars) are shown. The 1064 nm lidar ratio is given for the 1500 m retrieval interval length. The 1064 nm depolarization ratio was measured between 23:50 and 00:30 UTC. For more details, see Sect. 2.2.

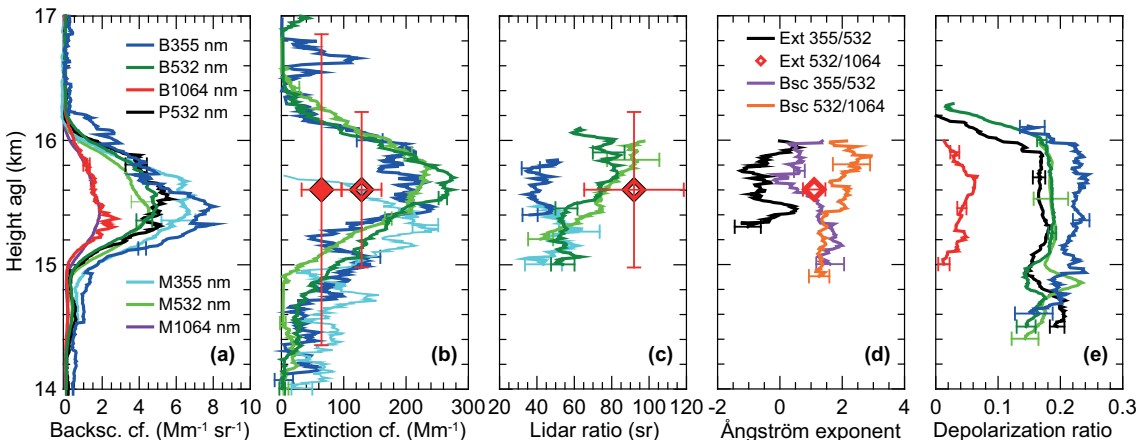

**Figure 3.** Same as Fig. 2, except for the stratospheric aerosol layer and for different signal smoothing lengths (as explained in Sect. 2.2). In the case of the 1064 nm extinction coefficient (red solid diamond) , a retrieval window length (least-squares method) of 2500 m had to be applied (indicated by the long vertical bar). We estimated the layer mean 1064 nm extinction coefficient (red open diamond) for the 1250 m thick layer from 15–16.25 km height by multiplying the obtain value for 2500 window length by a factor of 2 (see text for more details). In the subsequently calculation of the 1064 nm lidar ratio we used this 1250 m layer mean extinction value (open diamond) together with an appropriately smoothed backscatter coefficient (see text for more details). The most surprising finding is the strong difference between the tropospheric (Fig. 2e) and stratospheric (Fig. 3e) smoke depolarization ratios at 355 and 532 nm.

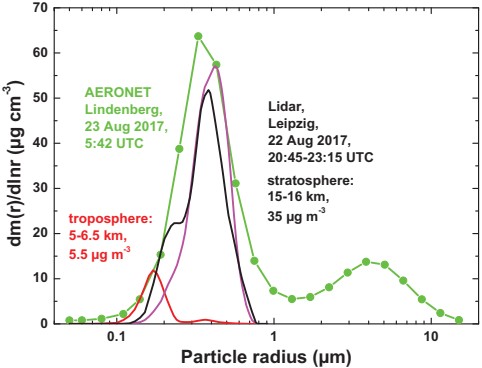

**Figure 4.** Particle mass size distribution derived from column (tropospheric + stratospheric) AERONET observations at Lindenberg, 180 km northeast of the lidar site, in the morning of 23 August 2017 (green) and obtained from the inversion of lidar-derived optical properties in the tropospheric layer (red) and stratospheric layer (black, magenta). Small particles prevailed in the tropospheric layer and comparably large accumulation-mode particles dominated in the stratospheric layer. The black and magenta curves are obtained by assuming spherical and spheroidal particle shapes in the lidar data inversion, respectively.

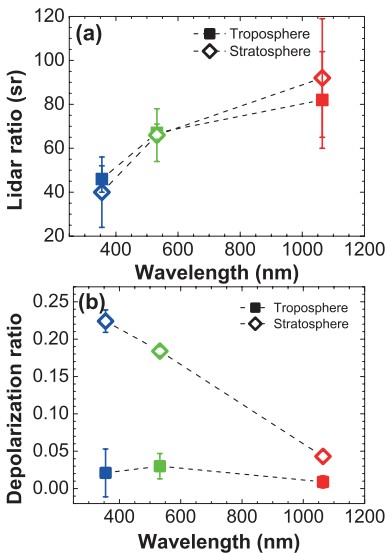

**Figure 5.** Comparison of the spectral dependence of the tropospheric (5-6 km height) and stratospheric (15-16 km height) particle lidar ratio (a) and particle linear depolarization ratio (b). A strongly contrasting spectral behavior is found in the case of the depolarization ratio and an almost similar wavelength dependence (in the troposphere and stratosphere) is found for the lidar ratio. Only BERTHA values are considered.

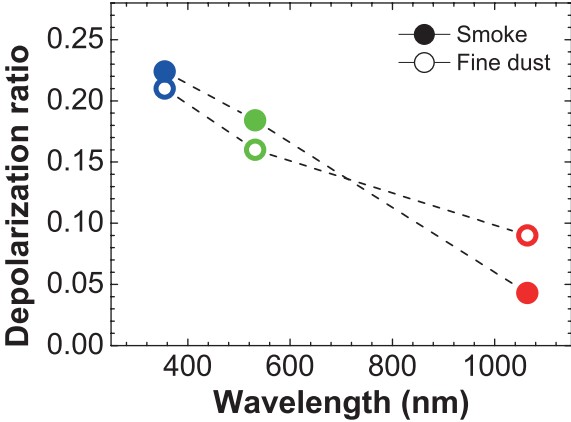

**Figure 6. NEW FIGURE! Spectral dependence of particle linear depolarization ratio for fine-mode desert dust and stratospheric accumulation-mode smoke particles.**