# Peer review of "Depolarization and lidar ratios at 355, 532, and 1064 nm and microphysical properties of aged tropospheric and stratospheric Canadian wildfire smoke"

_Atmospheric Chemistry and Physics, 2018_

## Referee Comment (RC1) · Anonymous Referee #1 · 30 May 2018

The authors present in their paper very interesting lidar observations during an exceptional smoke event captured above Europe during August 2017. They present measurements of the extinction coefficient, lidar ratio and particle depolarization ratio at three wavelengths, providing thus a unique dataset for characterizing the optical and microphysical properties of aged smoke. The highlight of the paper is the different depolarization ratio observed for the same smoke event in the troposphere and the stratosphere. These observation add new information to the database of optical and

microphysical properties of smoke particles which is extremely valuable for aerosol typing algorithms. The paper is suitable for publication in ACP and should be accepted after considering and clarifying few issues mentioned below:

General comment:

Although the paper is generally well written its structure leads to many repetitions. For instance there is a large paragraph in the introduction describing the three lidar systems and then a similar in length description is found in section 2.1. In the introduction in page 3 the authors comment in detail on the importance of their measurements before showing them. I guess that this part belongs to the discussion or the conclusions and possible outlook.

Similarly in section 3.2.1 the authors first summarize and interpret their findings before presenting them in more detail in 3.2 and 3.3 . The authors should reconsider the structure of section 3.

Specific comments:

Page 5, line 6: What do the authors consider as reasonably low uncertainty? What is the vertical resolution of the 1064nm extinction profiles when considering a regression window of 2500m?

Page 5, line 19: The authors use the method developed by Vesolovskii et al but provide reference for the theoretical background to Ansmann and Mueller 2005. Are there differences between them?

Page 6, lines 26-29: This description is confusing. What is "trustworthy estimate'? A figure would help to present the problem and its solution. As written it looks like an arbitrary adjustment to the extinction profile. How this adjustment would affect the estimated lidar ratio? Please comment.

---

## Referee Comment (RC2) · Anonymous Referee #2 · 17 Jul 2018

Review acp-2018-358:

Main comment: This manuscript presents the depolarization and microphysical properties of an extraordinary event of Canadian wildfire smoke detected in the stratosphere over central Europe. Being the second part of two papers about this event, the main findings are: 1) the quite complete information provided (lidar ratios and depolarizations at several wavelengths) and 2) the strange high values of the depolarization in

the stratosphere. Despite the reason of these high depolarization values remains unknown, authors provides consistent hypothesis. Thus, the presented work is a good contribution to the scientific community so I recommend its publication. I appreciate that it was easy to read and I didn't find any typo! However, the authors should consider the following comments:

Major comments: Errors in Table 1 should be included as performed in the rest of tables and graphics. This is quite important since the lack of these errors prevents to track the propagation error from optical to the microphysical properties, which leads me to my second major comment: microphysical property errors shown in Table 2 are around 30%, quite small to my opinion considering the assumptions and the way properties at 1064 nm are derived. I suggest including a detail explanation about the error propagation and its interpretation.

Minor comments:

- Page 6 line 1: How the PBL height was determined? Is the given value an average during the considered period? I guess that the PBL includes the residual layer, it would be nice if you can confirm.

- Page 8 line 30: I found Hu et al., 2018 or a very similar one in ACPD. Please, update the reference.

---

## Author Comment (AC1) · 31 Jul 2018

**Response to the Reviewers**

The authors are very grateful to the reviewers for their critical remarks and suggestions. Based on their input, the paper has been revised. The revised version of the paper is included in this letter (at the end).

Main changes are highlighted (bold) in the revised version.

A few general comments:

Point 1:

We submitted two papers with focus on the Canadian wildfires on 21-23 August 2017: Paper 1 (Ansmann et al., ACPD, revised version was submitted on 14 July 2018), and Paper 2 (this paper, Haarig et al.). Both papers look at the different aspects of the Canadian wildfire smoke.

The title of the revised version of paper 1 is: "Extreme levels of Canadian wildfire smoke in the stratosphere over central Europe on 21-22 August 2017".

Paper 1 provides an extended overview based on satellite observations (MODIS, OMI) from Canada to Europe, AERONET observations, and lidar profiling (only in terms of the 532 nm extinction coefficient and the aerosol optical depth) at three central European sites (Hohenpeissenberg near Munich, Leipzig, and Kosetice near Prague) during the main phase of the extreme smoke event. It includes a discussion on source identification for the smoke as well.

The title of the revised version of paper 2 (the title of the paper was also changed): "Depolarization and lidar ratios at 355, 532, and 1064 nm and microphysical properties of aged tropospheric and stratospheric Canadian wildfire smoke".

Paper 2 (this paper) concentrates on the specific optical, microphysical, and morphological (shape) properties of the smoke layers occurring simultaneously in the troposphere and in the stratosphere over Leipzig, Germany. The worldwide only triple-wavelength polarization/Raman lidar was used in this study.

Point 2:

We totally separated the contents of paper 1 and paper 2, to emphasize the stand-alone character of both papers. To that end, Figure 9 of the originally submitted paper 1 (submitted version, 4 April 2017) was moved to a revised paper 2 (now Figure 6 in the revised version).

Point 3:

As recommended by reviewer #1, the structure of paper 2 was significantly improved. This relates to redundancy and the result sections 3 and 4. Specifically, in our revised version, Sect. 3 contains the basic findings, Sect. 3.1 is an overview (of the case study), Sect. 3.2 shows the smoke optical properties, Sect. 3.3 presents the smoke microphysical properties. Furthermore, the discussion part is now better visible by introducing a new Sect. 4 (Discussion). This section contains the following subsections: Sect. 4.1 The unexpected wavelength dependence of the stratospheric depolarization ratio: an attempt of explanation (in the submitted version Sect. 3.3), Sect. 4.2: Smoke depolarization and lidar ratios: an updated literature review (in the submitted version Sect. 3.4). Main findings and conclusions are given in Sect.5.

**Item by item response to the comments of the reviewers (*our answers are in bold*):**

**Reviewer #1:**

The authors present in their paper very interesting lidar observations during an exceptional smoke event captured above Europe during August 2017. They present measurements of the extinction coefficient, lidar ratio and particle depolarization ratio at three wavelengths, providing thus a unique dataset for characterizing the optical and microphysical properties of aged smoke. The highlight of the paper is the different depolarization ratio observed for the same smoke event in the troposphere and the stratosphere. These observation add new information to the database of optical and microphysical properties of smoke particles which is extremely valuable for aerosol typing algorithms. The paper is suitable for publication in ACP and should be accepted after considering and clarifying few issues mentioned below:

General comment:

Although the paper is generally well written its structure leads to many repetitions. For instance there is a large paragraph in the introduction describing the three lidar systems and then a similar in length description is found in section 2.1. In the introduction in page 3 the authors comment in detail on the importance of their measurements before showing them. I guess that this part belongs to the discussion or the conclusions and possible outlook. Similarly in section 3.2.1 the authors first summarize and interpret their findings before presenting them in more detail in 3.2 and 3.3 . The authors should reconsider the structure of section 3.

***We agree and changed the structure and discussions accordingly. However, we leave motivating points for the paper (now page 2, 2-4 paragraphs) in Sect. 1, also to highlight the role of EARLINET in the efforts of aerosol characterization, aerosol typing and classification, and establishment of a vertically resolved aerosol climatology. We totally agree with the reviewer and changed the style of presentation and re-wrote sections 1 and 3. However, the facts in section 3 have not changed, so we do not highlight it (no text in bold, except one paragraph).***

Specific comments:

Page 5, line 6: What do the authors consider as reasonably low uncertainty? What is the vertical resolution of the 1064nm extinction profiles when considering a regression window of 2500m?

***The vertical resolution (= regression window length) is 2500m. This is too coarse to resolve profile structures in a smoke layer of just 1250 m depth. But unfortunately, we had no other chance. We had to collect the Raman signals over this height range of 2500 m to sample enough molecular backscatter signals to keep the signal-to-noise ratio at a tolerable level. At the end, the statistical uncertainty in the 1064 nm particle extinction coefficient was below 50%. This value of uncertainty is acceptable to our opinion.***

***In the revised version, we explain … in a better and more clear way … how we 'converted' the extinction values for the 2500m thick layer to the one for the smoke layer (1250m vertical extension). Because the aerosol optical thickness (AOT) remains the same for a given layer, disregarding the regression window length, the obtained AOT for the 2500 m regression window describes the AOT for the 1250 m deep layer. So, the layer mean extinction values (for the 1250 m layer) is just a factor of 2 higher than the layer mean extinction values for the 2500 m layer.***

***This is explained in Sect. 3.2 (paragraph in bold), when the respective Figure 3 is shown and discussed.***

Page 5, line 19: The authors use the method developed by Vesolovskii et al but provide reference for the theoretical background to Ansmann and Mueller 2005. Are there differences between them?

*We thank the reviewer for his/her very important comment. Yes there is a difference, Igor Veselovskii's method is more advanced. In our revised version, the text was changed accordingly (Sect. 2.3), and we do no longer mention the book chapter (Ansmann and Müller, 2005) to avoid confusion. This minor change is not indicated (not in bold) in the attached PDF.*

Page 6, lines 26-29: This description is confusing. What is "trustworthy estimate"? A figure would help to present the problem and its solution. As written it looks like an arbitrary adjustment to the extinction profile. How this adjustment would affect the estimated lidar ratio? Please comment.

*As we mentioned above, we changed the text (in Sect 2.2). 'Trustworthy estimate' is replaced by numbers: 10% uncertainty in the troposphere, and <50% in the stratosphere. This minor change is not indicated in the attached PDF.*

*We explain in Sect.3.2 in a more easy-to-follow and straight-forward way how we get the mean extinction coefficient for the smoke layer with a geometrical depth of 1250m. We used the 1250 m layer mean extinction coefficient to calculate the layer mean lidar ratio. We think that introducing a new figure will overload a paper and also shift the main findings of our paper to a different direction.*

**Reviewer #2:**

Main comment: This manuscript presents the depolarization and microphysical properties of an extraordinary event of Canadian wildfire smoke detected in the stratosphere over central Europe. Being the second part of two papers about this event, the main findings are: 1) the quite complete information provided (lidar ratios and depolarizations at several wavelengths) and 2) the strange high values of the depolarization in the stratosphere. Despite the reason of these high depolarization values remains unknown, authors provides consistent hypothesis. Thus, the presented work is a good contribution to the scientific community so I recommend its publication. I appreciate that it was easy to read and I didn't find any typo! However, the authors should consider the following comments:

Major comments:

Errors in Table 1 should be included as performed in the rest of tables and graphics. This is quite important since the lack of these errors prevents to track the propagation error from optical to the microphysical properties…

*We thank the reviewer for his/her careful reading. We added the (one standard deviation) uncertainty to the values in Table 1. The uncertainties are about 10-15%.*

.. which leads me to my second major comment: microphysical property errors shown in Table 2 are around 30%, quite small to my opinion considering the assumptions and the way properties at 1064 nm are derived. I suggest including a detail explanation about the error propagation and its interpretation.

*Uncertainties of 10-15% are included in the uncertainties in Table 2. This is now mentioned in Sect. 3.3 (in the Table 2 caption, not indicated in bold). The uncertainties in the inversion products are relatively small because the cases were 'quite' simple (easy to handle). The used 355, 532, and 1064 nm backscatter coefficients and 355 and 532 nm extinction coefficients (from Table 1) showed a clear structure and wavelength dependence. Note, that the 1064 nm extinction coefficient is not included in the present version of the inversion scheme as described in Sect. 2.3.*

Minor comments:

- Page 6 line 1: How the PBL height was determined? Is the given value an average during the considered period? I guess that the PBL includes the residual layer, it would be nice if you can confirm.

***In this work we do not explain how we determined the PBL, it is of minor importance in a paper mainly dealing with stratospheric smoke.  What we did:  We just checked the color plot for the obvious top height of the pollution layer. However, following the reviewer suggestion, we mention that we observed the residual layer and thus identified the top height of the residual …by the enhanced range-corrected signal  (see Sect. 3.1, not indicated in bold).***

- Page 8 line 30: I found Hu et al., 2018 or a very similar one in ACPD. Please, update the reference.

***We updated the reference and checked the paper and the observed depolarization ratios. We use the results for comparisons especially in Sect. 4.1 (Unexpected wavelength dependence of the stratospheric smoke depolarization ratio).***

[revised manuscript text omitted]